# INFORMATION-THEORETIC DIFFUSION

**Xianghao Kong**
University of California Riverside

**Rob Brekelmans**
Vector Institute

**Greg Ver Steeg**
University of California Riverside

## ABSTRACT

Denoising diffusion models have spurred significant gains in density modeling and image generation, precipitating an industrial revolution in text-guided AI art generation. We introduce a new mathematical foundation for diffusion models inspired by classic results in information theory that connect Information with Minimum Mean Square Error regression, the so-called I-MMSE relations. We generalize the I-MMSE relations to *exactly* relate the data distribution to an optimal denoising regression problem, leading to an elegant refinement of existing diffusion bounds. This new insight leads to several improvements for probability distribution estimation, including a theoretical justification for diffusion model ensembling. Remarkably, our framework shows how continuous and discrete probabilities can be learned with the same regression objective, avoiding domain-specific generative models used in variational methods. Code to reproduce experiments is provided at `https://github.com/kxh001/ITdiffusion` and simplified demonstration code is at `https://github.com/gregversteeg/InfoDiffusionSimple`.

## 1 INTRODUCTION

Denoising diffusion models (Sohl-Dickstein et al., 2015) incorporating recent improvements (Ho et al., 2020) now outperform GANs for image generation (Dhariwal & Nichol, 2021), and also lead to better density models than previously state-of-the-art autoregressive models (Kingma et al., 2021). The quality and flexibility of image results have led to major new industrial applications for automatically generating diverse and realistic images from open-ended text prompts (Ramesh et al., 2022; Saharia et al., 2022; Rombach et al., 2022). Mathematically, diffusion models can be understood in a variety of ways: as classic denoising autoencoders (Vincent, 2011) with multiple noise levels and a new architecture (Ho et al., 2020), as VAEs with a fixed noising encoder (Kingma et al., 2021), as annealed score matching models (Song & Ermon, 2019), as a non-equilibrium process that tractably bridges between a target distribution and a Gaussian (Sohl-Dickstein et al., 2015), or as a stochastic differential equation that does the same (Song et al., 2020; Liu et al., 2022).

In this paper, we call attention to a connection between diffusion models and a classic result in information theory relating the mutual information to the Minimum Mean Square Error (MMSE) estimator for denoising a Gaussian noise channel (Guo et al., 2005). Research on Information and MMSE (often referred to as I-MMSE relations) transformed information theory with new representations of standard measures leading to elegant proofs of fundamental results (Verdú & Guo, 2006). This paper uses a generalization of the I-MMSE relation to discover an exact relation between data probability distribution and optimal denoising regression. The information-theoretic formulation of diffusion simplifies and improves existing results. Our main contributions are as follows.

- A new, exact relation between probability density and the global optimum of a mean square error denoising objective of the form, $-\log p(\boldsymbol{x}) = 1/2 \int_0^\infty \mathrm{mmse}(\boldsymbol{x}, \gamma) d\gamma + \text{constant terms}$, with useful variations in Eq. (8), (9) and (11). Because this expression is exact and not a bound, we can convert any statements about density functionals such as entropy into statements about the unconstrained optimum of regression problems easily solved with neural nets.
- The same optimization problem can also be exactly related to *discrete* probability mass, Eq. (12). In contrast, the variational diffusion bound requires specifying a separate discrete decoder term in the generative model and associated variational bound. Our unification of discrete and continuous probabilities justifies empirical work applying diffusion to categorical variables.

- In experiments, we show that our approach can take pre-trained discrete diffusion models and re-interpret them as continuous density models with competitive log-likelihoods. Our approach allows us to fine-tune and *ensemble* existing diffusion models to achieve better NLLs.

## 2 FUNDAMENTAL POINTWISE DENOISING RELATION

Let $p(\boldsymbol{z}_\gamma|\boldsymbol{x})$ be a Gaussian noise channel with $\boldsymbol{z}_\gamma = \sqrt{\gamma}\boldsymbol{x} + \boldsymbol{\epsilon}$ and $\boldsymbol{\epsilon} \sim \mathcal{N}(0, \mathbb{I})$, where $\gamma$ represents the Signal-to-Noise Ratio (SNR) and $p(\boldsymbol{x})$ is the unknown data distribution. This channel has exploding variance as the SNR increases but we will see that the variance of this channel can be normalized arbitrarily without affecting results. Our convention matches the information theory literature and significantly simplifies proofs.

The seminal result of Guo et al. (2005) connects mutual information with MMSE estimators,

$$\frac{d}{d\gamma}I(\boldsymbol{x};\boldsymbol{z}_\gamma) = \text{\small 1/2}\,\text{mmse}(\gamma).$$ (1)

Here, the MMSE refers to the Minimum Mean Square Error for recovering $\boldsymbol{x}$ in this noisy channel,

$$\text{mmse}(\gamma) \equiv \min_{\hat{\boldsymbol{x}}(\boldsymbol{z}_\gamma,\gamma)} \mathbb{E}_{p(\boldsymbol{z}_\gamma,\boldsymbol{x})}\big[\|\boldsymbol{x} - \hat{\boldsymbol{x}}(\boldsymbol{z}_\gamma,\gamma)\|_2^2\big].$$ (2)

We refer to $\hat{\boldsymbol{x}}$ as the denoising function. The optimal denoising function $\hat{\boldsymbol{x}}^*$ corresponds to the conditional expectation, which can be seen using variational calculus or from the fact that the squared error is a Bregman divergence ( Banerjee et al. (2005) Prop. 1),

$$\hat{\boldsymbol{x}}^*(\boldsymbol{z}_\gamma,\gamma) \equiv \underset{\hat{\boldsymbol{x}}(\boldsymbol{z}_\gamma,\gamma)}{\arg\min}\,\text{mse}(\gamma) = \mathbb{E}_{\boldsymbol{x}\sim p(\boldsymbol{x}|\boldsymbol{z}_\gamma)}[\boldsymbol{x}]$$ (3)

The analytic solution is typically intractable because it requires sampling from the posterior distribution of the noise channel.

Recent work on variational diffusion models writes a lower bound on log likelihood explicitly in terms of MMSE (Kingma et al., 2021), suggesting a potential connection to Eq. (1). However, the precise nature of the connection is not clear because the variational diffusion bound is an inequality while Eq. (2) is an equality, and the variational bound is formulated pointwise for $\log p(\boldsymbol{x})$ at a single $\boldsymbol{x}$, while Eq. (2) is an expectation.

We now introduce a point-wise generalization of Guo et al's result that is the foundation for all other new results in this paper.

$$\boxed{\frac{d}{d\gamma}D_{KL}[p(\boldsymbol{z}_\gamma|\boldsymbol{x}) \parallel p(\boldsymbol{z}_\gamma)] = \text{\small 1/2}\,\text{mmse}(\boldsymbol{x},\gamma)}$$ (4)

The marginal is $p(\boldsymbol{z}_\gamma) = \int p(\boldsymbol{z}_\gamma|\boldsymbol{x})p(\boldsymbol{x})d\boldsymbol{x}$, and the pointwise MMSE is defined as follows,

$$\text{mmse}(\boldsymbol{x},\gamma) \equiv \mathbb{E}_{p(\boldsymbol{z}_\gamma|\boldsymbol{x})}\big[\|\boldsymbol{x} - \hat{\boldsymbol{x}}^*(\boldsymbol{z}_\gamma,\gamma)\|_2^2\big].$$ (5)

Pointwise MMSE is just the MMSE evaluated at a single point $\boldsymbol{x}$, and $\mathbb{E}_{p(\boldsymbol{x})}[\text{mmse}(\boldsymbol{x},\gamma)] = \text{mmse}(\gamma)$. Taking the expectation with respect to $\boldsymbol{x}$ of both sides of Eq. (4) recovers Guo et al's famous result, Eq. (1). Our proof of Eq. (4) uses special properties of the Gaussian noise channel and repeated application of integration by parts, with a detailed proof given in Appendix A.1. This result can also be seen as a special case of Theorem 5 from Palomar & Verdú (2005) by replacing their general channel with our channel written in terms of $\gamma$ and then using the chain rule. In the rest of this paper, we show how this more general version of the I-MMSE relation can be used to reformulate denoising diffusion models.

## 3 DIFFUSION AS THERMODYNAMIC INTEGRATION

We first use Eq. (4) to derive an expression for log-likelihood that resembles the variational bound. However, using results from the information theory literature (Verdú & Guo, 2006), we find that the expression can be significantly simplified and certain terms that are typically estimated can be calculated analytically.

Our derivation is inspired by the recent development of thermodynamic variational inference (Masrani et al., 2019; Brekelmans et al., 2020) which shows how to construct a path connecting a tractable distribution to some target, such that integrating over this path recovers the log likelihood for the target model. The paths between distributions can be generalized in a number of ways (Masrani et al., 2021; Chen et al., 2021). Typically, though, these path integrals are difficult to estimate because they require expensive sampling from complex intermediate distributions. A distinctive property of diffusion models is that the Gaussian noise channel, which transforms the target distribution into a standard normal distribution, can be easily sampled at any intermediate noise level.

We use the fundamental theorem of calculus to evaluate a function at two points in terms of the integral of its derivative, $\int_{\gamma_0}^{\gamma_1} d\gamma \frac{d}{d\gamma} f(\gamma) = f(\gamma_1) - f(\gamma_0)$. This approach is known as "thermodynamic integration" in the statistical physics literature (Ogata, 1989; Gelman & Meng, 1998), where evaluation of the endpoints corresponds to a difference in the free energy or log partition function, and the derivatives of these quantities may be more amenable to Monte Carlo simulation.

Consider applying the thermodynamic integration trick to the following function, $f(\boldsymbol{x}, \gamma) \equiv D_{KL}[p(\boldsymbol{z}_\gamma|\boldsymbol{x}) \parallel p(\boldsymbol{z}_\gamma)]$. As before, we have $p(\boldsymbol{x})$, the data distribution, and a Gaussian noise channel, $\boldsymbol{z}_\gamma = \sqrt{\gamma}\boldsymbol{x} + \boldsymbol{\epsilon}$ with different signal-to-noise ratios $\gamma$,

$$\int_{\gamma_0}^{\gamma_1} d\gamma \frac{d}{d\gamma} f(\boldsymbol{x}, \gamma) = D_{KL}[p(\boldsymbol{z}_{\gamma_1}|\boldsymbol{x}) \parallel p(\boldsymbol{z}_{\gamma_1})] - D_{KL}[p(\boldsymbol{z}_{\gamma_0}|\boldsymbol{x}) \parallel p(\boldsymbol{z}_{\gamma_0})]$$

$$= D_{KL}[p(\boldsymbol{z}_{\gamma_1}|\boldsymbol{x}) \parallel p(\boldsymbol{z}_{\gamma_1})] - \mathbb{E}_{p(\boldsymbol{z}_{\gamma_0}|\boldsymbol{x})}[\log p(\boldsymbol{x}|\boldsymbol{z}_{\gamma_0})] + \log p(\boldsymbol{x}).$$

In the second line, we expanded the KL divergence and used Bayes rule. Next, we re-arrange and use Eq. (4) to re-write the integrand

$$-\log p(\boldsymbol{x}) = \underbrace{D_{KL}[p(\boldsymbol{z}_{\gamma_1}|\boldsymbol{x}) \parallel p(\boldsymbol{z}_{\gamma_1})]}_{\text{Prior loss}} + \underbrace{\mathbb{E}_{p(\boldsymbol{z}_{\gamma_0}|\boldsymbol{x})}[-\log p(\boldsymbol{x}|\boldsymbol{z}_{\gamma_0})]}_{\text{Reconstruction loss}} - \underbrace{\tfrac{1}{2}\int_{\gamma_0}^{\gamma_1} \text{mmse}(\boldsymbol{x}, \gamma)d\gamma}_{\text{Diffusion loss}}. \quad (6)$$

Comparing to a particular variational bound for diffusion models in the continuous time limit (Eq. 15 from Kingma et al. (2021)), we see this expression looks similar (see App. A.7 for more details). However, our derivation so far is exact and we haven't introduced any variational approximations. Prior and reconstruction loss terms are stochastically estimated in the variational formulation, but we show this is unwise and unnecessary. Consider the limit where $\gamma_1 \to 0$. In that case, the prior loss will be zero. The reconstruction term, for continuous densities, becomes infinite in the limit of large $\gamma_0$. For this reason, recent diffusion models only consider reconstruction for discrete random variables, $P(\boldsymbol{x}|\boldsymbol{z}_{\gamma_0})$. Estimation of conflicting divergent terms and the need for separate prior and reconstruction objectives can be avoided, as we now show.

**Simple and exact probability density via MMSE** We consider applying thermodynamic integration to a slightly different function, and expand the range of integration to $\gamma \in [0, \infty)$. Consider sending samples from either the data distribution $p(\boldsymbol{x})$ or a standard Gaussian $p_G(\boldsymbol{x}) = \mathcal{N}(\boldsymbol{x}; 0, \mathbb{I})$ through our Gaussian noise channel. We denote the MMSE for the channel with Gaussian input as $\text{mmse}_G(\gamma)$, and write its marginal output distribution as $p_G(\boldsymbol{z}_\gamma) = \int p(\boldsymbol{z}_\gamma|\boldsymbol{x})p_G(\boldsymbol{x})d\boldsymbol{x}$. Finally, we define the function $f(\boldsymbol{x}, \gamma)$ as [1]

$$f(\boldsymbol{x}, \gamma) \equiv D_{KL}[p(\boldsymbol{z}_\gamma|\boldsymbol{x}) \parallel p_G(\boldsymbol{z}_\gamma)] - D_{KL}[p(\boldsymbol{z}_\gamma|\boldsymbol{x}) \parallel p(\boldsymbol{z}_\gamma)].$$

In the limit of zero SNR, we get $\lim_{\gamma \to 0} f(\boldsymbol{x}, \gamma) = 0$. In the high SNR limit we use the following result proved in App. A.2,

$$\lim_{\gamma \to \infty} f(\boldsymbol{x}, \gamma) = \log \frac{p(\boldsymbol{x})}{p_G(\boldsymbol{x})}. \quad (7)$$

Combining this with Eq. (4), we can write the log likelihood *exactly* in terms of the log likelihood of a Gaussian and a one dimensional integral.

$$-\log p(\boldsymbol{x}) = -\log p_G(\boldsymbol{x}) - \int_0^\infty d\gamma \frac{d}{d\gamma} f(\boldsymbol{x}, \gamma)$$

$$= -\log p_G(\boldsymbol{x}) - \tfrac{1}{2}\int_0^\infty d\gamma \left(\text{mmse}_G(\boldsymbol{x}, \gamma) - \text{mmse}(\boldsymbol{x}, \gamma)\right) \quad (8)$$

---

[1] Note that $\mathbb{E}_{p(\boldsymbol{x})}[f(\boldsymbol{x}, \gamma)] = \mathbb{E}_{p(\boldsymbol{x}, \boldsymbol{z}_\gamma)}[\log p(\boldsymbol{z}_\gamma) - \log p_G(\boldsymbol{z}_\gamma)] = D_{KL}[p(\boldsymbol{z}_\gamma) \parallel p_G(\boldsymbol{z}_\gamma)]$ is the gap in an upper bound $\mathbb{E}_{p(\boldsymbol{x})}[D_{KL}[p(\boldsymbol{z}_\gamma|\boldsymbol{x}) \parallel p_G(\boldsymbol{z}_\gamma)]]$ on mutual information $I(\boldsymbol{x}; \boldsymbol{z}_\gamma)$, where the upper bound uses the marginal distribution induced by the Gaussian source $p_G(\boldsymbol{z}_\gamma)$ instead of the data $p(\boldsymbol{z}_\gamma)$.

This expresses density in terms of a Gaussian density and a correction that measures how much better we can denoise the target distribution than we could using the optimal decoder for Gaussian source data. The density can be further simplified by writing out the Gaussian expressions explicitly and simplifying with an identity given in App. A.3.

$$-\log p(\boldsymbol{x}) = d/2 \log(2\pi e) - {}^{1}\!/{}_{2} \int_0^\infty d\gamma \left( \frac{d}{1+\gamma} - \mathrm{mmse}(\boldsymbol{x}, \gamma) \right) \tag{9}$$

This expression shows that density can be written solely in terms of the global optimum of a particular regression problem, the denoising MSE. This is convenient because neural networks excel at unconstrained optimization of MSE loss functions. If we take the expectation of $-\log p(\boldsymbol{x})$ using Eq. (9), we recover a relatively recently discovered representation of the differential entropy, $h(p)$, in terms of MMSE (Verdú & Guo, 2006). This highlights an advantage of our approach, as all density functionals can be rewritten in terms of the solution of an unconstrained regression problem,

$$h(p) \equiv \mathbb{E}_{p(\boldsymbol{x})}[-\log p(\boldsymbol{x})] = d/2 \log 2\pi e - {}^{1}\!/{}_{2} \int_0^\infty d\gamma \left( \frac{d}{1+\gamma} - \mathrm{mmse}(\gamma) \right). \tag{10}$$

Since the first integral in Eq. (9)-(10) does not depend on $\boldsymbol{x}$, it is tempting to absorb it into a constant. However, the first integral diverges, and the second integral typically divergences as well – only the difference converges. This observation will help improve density estimation, by noticing that only the gap between MMSEs is important and that the gap becomes small at high and low values of SNR.

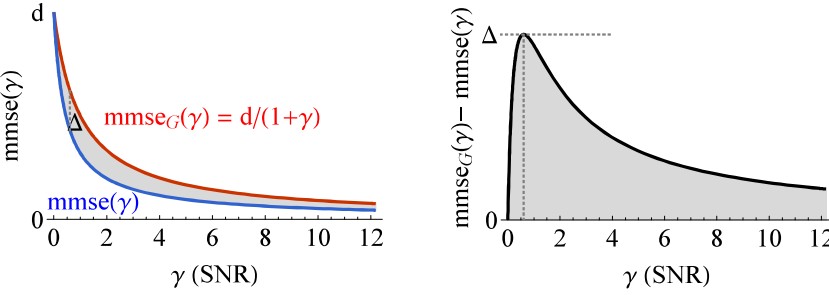

Figure 1: The integral of the gap between MMSE curves for data from the target distribution versus data from a Gaussian distribution is used in Eq. (10) to get an exact expression for the entropy, or expected Negative Log Likelihood (NLL), of the data.

We show example MMSE curves for denoising with Gaussian input versus non-Gaussian inputs in Fig. 1. The goal is to integrate the gap between these two curves, one analytic (for Gaussians) and one given by the data. Note that the gap could in principle be positive or negative. However, we know that Gaussians have the maximum possible MMSE, compared to any distribution with the same variance or covariance, when used as an input to a Gaussian noise channel (Shamai & Verdu, 1992). Therefore, if we always compare to Gaussians with the same variance or covariance as the data, we can guarantee that the gap is positive. In practice, if the gap ever becomes negative, which we show can occur for models appearing in the literature in Sec. 5, it means that the discovered estimator is sub-optimal and we can fall back to the optimal Gaussian estimator to improve results in this region.

We also see that for large and small SNR values, the gap becomes small. At low SNR, the noisy data is approximately Gaussian. At high SNR, the noise goes to zero, so a linear (Gaussian) denoiser is sufficient in either case. Recognizing that the important signals come from intermediate SNR values can help save computation over approaches that indiscriminately optimize over the whole curve, as we discuss in Sec. 4.

Standardizing data to have unit variance to ensure that the MMSE gap is always positive may be inconvenient, so we derive a variation of Eq. (9) that takes the scale into account. Instead of using $p_G(\boldsymbol{x}) = \mathcal{N}(\boldsymbol{x}; 0, \mathbb{I})$, we can take the base measure to be a Gaussian, $p_G(\boldsymbol{x}) = \mathcal{N}(\boldsymbol{x}; \boldsymbol{\mu}, \boldsymbol{\Sigma})$, with mean and covariance that match the data. Letting $\lambda_1, \ldots, \lambda_d$ be the eigenvalues of the covariance

matrix $\boldsymbol{\Sigma}$, we derive a compact expression in App. A.4,

$$\mathbb{E}_{p(\boldsymbol{x})}[-\log p(\boldsymbol{x})] = \underbrace{1/2 \log \det(2\pi e \boldsymbol{\Sigma})}_{\text{Gaussian entropy}} - \underbrace{1/2 \int_0^\infty d\gamma \left( \sum_{i=1}^d \frac{1}{\gamma + 1/\lambda_i} - \text{mmse}(\gamma) \right)}_{\text{Deviation from Gaussianity} \geq 0}. \quad (11)$$

This expression tells us precisely which part of the reconstruction mean square error curve is actually important, namely the part that deviates most from Gaussianity. Note that if the eigenvalues of $\boldsymbol{\Sigma}$ are not feasible to estimate, we can use a diagonal covariance matrix for the base Gaussian and still preserve the desired property that the gap between the data MMSE and Gaussian MMSE is non-negative (Shamai & Verdu, 1992).

Finally, we make several remarks to relate our exact expression for the likelihood in Eq. (9) to existing work in the diffusion literature.

- Since the MSE minimization is performed at each $\gamma$, reweighting terms with different $\gamma$ (Kingma et al., 2021) should not have an effect as long as we achieve the global minima at each $\gamma$.
- Several heuristics have been suggested for improving the "noise schedule", which corresponds to points for evaluating the MMSE integral in our formulation (Ho et al., 2020; Kingma et al., 2021; Nichol & Dhariwal, 2021). Our formulation highlights the importance of sampling from SNRs where there is a large gap, and we suggest a simple and effective strategy for this in Sec. 4.
- The literature on diffusion models also suggests more complex denoising distributions that also model the covariance (Nichol & Dhariwal, 2021). Modeling the covariance of the denoising model is unnecessary in our approach, as our exact expressions never require it.
- Compare Eq. (9) to the exact, continuous density expression in Eq. 39 of Song et al. (2020), which combines neural ODE flows and diffusion models. The form of that expression is:

$$-\log p(\boldsymbol{x}(0)) = -\log p_G(\boldsymbol{x}(T)) + \int_0^T \nabla \cdot \boldsymbol{g}(\boldsymbol{x}(t), t) dt.$$

  The first step to using this expression is to solve a differential equation for the full trajectory, $\boldsymbol{x}(t)$, that depends on many evaluations of a learned denoising diffusion (or score) model. Then, estimating the integral is highly non-trivial as $\boldsymbol{g}$ is complex and its divergence is expensive to compute, necessitating some stochastic approximations. See Sec. 6 for additional discussion.

**Discrete probability estimator** Modeling probability distributions over continuous versus discrete random variables typically requires rather different approaches. An unusual feature of I-MMSE relations is that they naturally handle both types of random variables. In this subsection, consider that $\boldsymbol{x} \in \mathcal{X} \subset \mathcal{Z}^d$, with a domain that is discrete but numeric (non-numeric discrete variables can be handled via an appropriate mapping function (Guo et al., 2005)). We will use capital $P(\boldsymbol{x})$ to denote probability (rather than probability *density*) over this discrete domain. Note that the Gaussian noise channel, $\boldsymbol{z}_\gamma = \sqrt{\gamma}\boldsymbol{x} + \epsilon$, is still continuous with an associated probability density, $p(\boldsymbol{z}_\gamma|\boldsymbol{x})$.

Re-doing the analysis in Eq. (6) leads to the same expression, but replacing $p(\boldsymbol{x})$ with $P(\boldsymbol{x})$. The first two terms in Eq. (6) go to zero,

$$\lim_{\gamma_0 \to \infty, \gamma_1 \to 0} D_{KL}[p(\boldsymbol{z}_{\gamma_1}|\boldsymbol{x}) \| p(\boldsymbol{z}_{\gamma_1})] + \mathbb{E}_{p(\boldsymbol{z}_{\gamma_0}|\boldsymbol{x})}[-\log P(\boldsymbol{x}|\boldsymbol{z}_{\gamma_0})] = 0.$$

This leads to the following form for the discrete probability.

$$\boxed{-\log P(\boldsymbol{x}) = 1/2 \int_0^\infty \text{mmse}(\boldsymbol{x}, \gamma) d\gamma} \quad (12)$$

At large SNR, $\boldsymbol{z}_\gamma$ is very concentrated around a discrete point specified by $\boldsymbol{x}$. In that case, we should be able to recover the true value from the slightly perturbed value with high probability. Sec. 4 derives tail bounds used for approximating this integral numerically. Taking the expectation of this general result recovers an equation for the Shannon entropy from Section VI of Guo et al. (2005),

$$H(\boldsymbol{x}) \equiv -\sum_{\boldsymbol{x} \in \mathcal{X}} P(\boldsymbol{x}) \log P(\boldsymbol{x}) = \mathbb{E}_{\boldsymbol{x} \sim P(\boldsymbol{x})}[-\log P(\boldsymbol{x})] = 1/2 \int_0^\infty \text{mmse}(\gamma) d\gamma. \quad (13)$$

Conveniently, our discrete and continuous probability representations are nearly identical – they rely on the same integral and optimization and differ only by constant terms. We discuss comparisons to the variational bounds in App. A.7 and numerical implementation details in Sec. 4.

## 4 IMPLEMENTATION

While we start with exact expressions for density and discrete probability in Eqs. (9) and (12), in practice there will be two sources of error when implementing our approach numerically. First, we must parametrize our estimator as a neural network that may not achieve the global minimum mean square error and, second, we are forced to rely on numerical integration.

**MMSE Upper Bounds**     While the likelihood bounds in Eqs. (6), (8), and (9) are exact, evaluating each $\mathrm{mmse}(\boldsymbol{x}, \gamma)$ term requires access to the optimal denoising function or conditional expectation $\hat{\boldsymbol{x}}^*(\boldsymbol{z}_\gamma, \gamma) = \mathbb{E}_{p(\boldsymbol{x}|\boldsymbol{z}_\gamma)}[\boldsymbol{x}]$. Using a suboptimal denoising function $\hat{\boldsymbol{x}}(\boldsymbol{z}_\gamma, \gamma)$ instead of $\hat{\boldsymbol{x}}^*(\boldsymbol{z}_\gamma, \gamma)$ to approximate the MMSE, we obtain an upper bound whose gap can be characterized as

$$\mathbb{E}_{p(\boldsymbol{x},\boldsymbol{z}_\gamma)}\big[\|\boldsymbol{x}-\hat{\boldsymbol{x}}(\boldsymbol{z}_\gamma,\gamma)\|_2^2\big] = \underbrace{\mathbb{E}_{p(\boldsymbol{x},\boldsymbol{z}_\gamma)}\big[\|\boldsymbol{x}-\hat{\boldsymbol{x}}^*(\boldsymbol{z}_\gamma,\gamma)\|_2^2\big]}_{\mathrm{mmse}(\gamma)} + \underbrace{\mathbb{E}_{p(\boldsymbol{z}_\gamma)}\big[\|\hat{\boldsymbol{x}}^*(\boldsymbol{z}_\gamma,\gamma)-\hat{\boldsymbol{x}}(\boldsymbol{z}_\gamma,\gamma)\|_2^2\big]}_{\text{estimation gap}}.$$

In App. A.6, we derive this upper bound and its gap using results of Banerjee et al. (2005) for general Bregman divergences. This translates to an upper bound on the Negative Log Likelihood (NLL),

$$\mathbb{E}_{p(\boldsymbol{x})}\big[-\log p(\boldsymbol{x})\big] \leq 1/2 \log\det(2\pi e \boldsymbol{\Sigma}) - 1/2 \int_0^\infty d\gamma \left( \sum_{i=1}^d \frac{1}{\gamma + 1/\lambda_i} - \mathbb{E}_{p(\boldsymbol{x},\boldsymbol{z}_\gamma)}\big[\|\boldsymbol{x}-\hat{\boldsymbol{x}}(\boldsymbol{z}_\gamma,\gamma)\|_2^2\big] \right).$$

**Restricting Range of Integration**     As noted in Fig. 1, only the difference $\mathrm{mmse}_G(\gamma) - \mathrm{mmse}(\gamma) \geq 0$ contributes to our expected NLL bound. The non-negativity of this difference is guaranteed by the results of Shamai & Verdu (1992), but may not hold in practice due to our suboptimal estimators of the MMSE, $\mathbb{E}_{p(\boldsymbol{x},\boldsymbol{z}_\gamma)}[\|\boldsymbol{x}-\hat{\boldsymbol{x}}(\boldsymbol{z}_\gamma,\gamma)\|_2^2] \geq \mathrm{mmse}(\gamma)$.

In regions $\gamma \leq \gamma_0$ or $\gamma \geq \gamma_1$ where the difference in mean square error terms appears to be negative, we can simply define $\hat{\boldsymbol{x}}(\boldsymbol{z}_\gamma, \gamma) \equiv \hat{\boldsymbol{x}}_G^*(\boldsymbol{z}_\gamma, \gamma)$, where $\hat{\boldsymbol{x}}_G^*(\boldsymbol{z}_\gamma, \gamma)$ is the optimal, linear decoder for a Gaussian with the same covariance as the data (App. A.5). For this choice of decoder, the integrand becomes zero so that we may drop the tails of the integral outside of the appropriate range $\gamma \in [\gamma_0, \gamma_1]$,

$$\mathbb{E}_{p(\boldsymbol{x})}\big[-\log p(\boldsymbol{x})\big] \leq 1/2 \log\det(2\pi e \boldsymbol{\Sigma}) - 1/2 \int_{\gamma_0}^{\gamma_1} d\gamma \left( \sum_{i=1}^d \frac{1}{\gamma + 1/\lambda_i} - \mathbb{E}_{p(\boldsymbol{x},\boldsymbol{z}_\gamma)}[\|\boldsymbol{x}-\hat{\boldsymbol{x}}(\boldsymbol{z}_\gamma,\gamma)\|_2^2] \right).$$

**Parametrization**     We parametrize $\hat{\boldsymbol{x}}(\boldsymbol{z}_\gamma, \gamma) \equiv (\boldsymbol{z}_\gamma - \hat{\epsilon}(\boldsymbol{z}_\gamma, \gamma))/\sqrt{\gamma}, \forall \gamma \in [\gamma_0, \gamma_1]$. With this definition, we can re-arrange to see that the error for reconstructing the noise, $\epsilon$, is related to the error for reconstructing the original image, $\hat{\epsilon}(\boldsymbol{z}_\gamma, \gamma) - \epsilon = \sqrt{\gamma}(\boldsymbol{x} - \hat{\boldsymbol{x}}(\boldsymbol{z}_\gamma, \gamma))$. Our neural network then implements $\hat{\epsilon}$, a function that predicts the noise. The inputs to this network are $(\boldsymbol{z}_\gamma, \gamma)$. Note that we can equivalently parametrize our network to use the inputs $(\boldsymbol{z}_\gamma/\sqrt{1+\gamma}, \gamma)$. This choice is preferable so that the variance of the noisy image representation is bounded, while the SNR is unchanged. Numerically, it is better to work with log SNR values, so we change variables, $\alpha = \log \gamma$, which leads to the following form

$$\mathbb{E}_{p(\boldsymbol{x})}\big[-\log p(\boldsymbol{x})\big] \leq 1/2 \log\det(2\pi e \boldsymbol{\Sigma}) - 1/2 \int_{\alpha_0}^{\alpha_1} d\alpha \left( f_\Sigma(\alpha) - \mathbb{E}_{\boldsymbol{x},\epsilon}\big[\|\epsilon - \hat{\epsilon}(\boldsymbol{z}_\gamma,\gamma)\|_2^2\big] \right). \quad (14)$$

Here, $f_\Sigma(\alpha) \equiv \sum_{i=1}^d \sigma(\alpha + \log \lambda_i)$, using the traditional sigmoid function, $\sigma(t) = 1/(1 + e^{-t})$.

**Numerical Integration**     The last technical issue to solve is how to evaluate the integral in Eq. (14) numerically. We use importance sampling to write this expression as an expectation over some distribution, $q(\alpha)$, for which we can get unbiased estimates via Monte Carlo sampling. This leads to the final specification of our loss function $\mathbb{E}_{p(\boldsymbol{x})}[-\log p(\boldsymbol{x})] \leq \mathcal{L}$, where

$$\mathcal{L} \equiv 1/2 \log\det(2\pi e \boldsymbol{\Sigma}) - 1/2 \mathbb{E}_{q(\alpha)}\big[1/q(\alpha) \big( f_\Sigma(\alpha) - \mathbb{E}_{\boldsymbol{x},\epsilon}\big[\|\epsilon - \hat{\epsilon}(\boldsymbol{z}_\gamma,\gamma)\|_2^2\big]\big)\big]. \quad (15)$$

All that remains is to choose $q(\alpha)$. We use our analysis of the Gaussian case to motivate this choice. Note that $f_\Sigma(\alpha)$ is a mixture of CDFs for logistic distributions with unit scale and different means. Mixtures of logistics with different means are well approximated by a logistic distribution with a larger scale (Crooks, 2009). So we take $q(\alpha)$ to be a truncated logistic distribution with mean $\mu = \mathrm{Mean}_i(-\log \lambda_i)$ and scale $s = \sqrt{1 + 3/\pi^2 \mathrm{Var}_i(\log \lambda_i)}$, based on moment matching to the

mixture of logistics implied by $f_\Sigma(\alpha)$. Samples can be generated from a uniform distribution using the quantile function, $\alpha = \mu + s \log t/(1-t), t \sim \mathcal{U}[t_0, t_1]$. The quantile range is set so that $\alpha \in [\mu - 4s, \mu + 4s]$. The objective is estimated with Monte Carlo sampling, providing a stochastic, unbiased estimate of our upper bound in Eq. (14). Numerical integration alternatives include the trapezoid rule (Lartillot & Philippe, 2006; Friel et al., 2014; Hug et al., 2016) or a Riemann sum, due to the monotonicity of $\mathrm{mmse}(\boldsymbol{x}, \gamma)$ (Kingma et al. (2021) Fig. 2, Masrani et al. (2019)).

**Comparing between continuous and discrete probability estimators**     Recent diffusion models are trained assuming discrete data. We can measure how well they model continuous data by viewing continuous density estimation as the limiting density of discrete points (Jaynes, 2003). Treating $p(\boldsymbol{x})$ as a uniform density in some $d$-dimensional box or bin of volume $\Delta^d$ around each discrete point leads to the relation, $\mathbb{E}[-\log p(\boldsymbol{x})] = \mathbb{E}[-\log P(\boldsymbol{x} \in \text{ bin})/(\Delta)^d] = \mathbb{E}[-\log P(\boldsymbol{x} \in \text{ bin}) - d \log \Delta]$. In the other direction, when we use a continuous density estimator to model a discrete density, we use uniform dequantization (Ho et al., 2019).

**Direct discrete probability estimate**     Finally, we derive a practical upper bound on negative log likelihood for discrete probabilities.

$$\mathbb{E}[-\log P(\boldsymbol{x})] = 1/2 \int_0^\infty \mathrm{mmse}(\gamma)d\gamma = 1/2 \int_{\gamma_0}^{\gamma_1} \mathrm{mmse}(\gamma)d\gamma + \textcolor{blue}{1/2 \left( \int_0^{\gamma_0} + \int_{\gamma_1}^\infty \right) \mathrm{mmse}(\gamma)d\gamma}$$

$$\mathbb{E}[-\log P(\boldsymbol{x})] \leq 1/2 \int_{\gamma_0}^{\gamma_1} \mathbb{E}_{\boldsymbol{z}_\gamma, \boldsymbol{x}}\left[ \|\boldsymbol{x} - \hat{\boldsymbol{x}}(\boldsymbol{z}_\gamma, \gamma)\|_2^2 \right] d\gamma + \textcolor{blue}{c(\gamma_0, \gamma_1)} \tag{16}$$

In App. B.1, we analytically derive upper bounds for the left and right tail of the integral, expressed using $c(\gamma_0, \gamma_1)$. We also get an upper bound from using a denoiser that is not necessarily globally optimal. The discrete and continuous density estimators differ only by constants, therefore we can use the same importance sampling estimator for the integral, as in Eq. 15.

# 5 EXPERIMENTS

Our results establish an exact connection between the data likelihood and the solution to a regression problem, the Gaussian denoising MMSE. If we integrate the MSE curve of a particular denoiser and get a worse bound, it simply means that the denoiser does not achieve the MMSE. Interestingly, variational diffusion models optimize an objective that can be made quite similar with appropriate choices for variational distributions (App. A.7). Therefore, we can take previously trained diffusion models and evaluate alternate likelihood bounds using the methods described above. We can attempt to improve these bounds by directly optimizing the denoising MSE. Finally, a straightforward implication of our results is that we would like the minimum MSE denoiser at each SNR value; therefore, if we have different denoisers with lower error at different SNR values, we can combine them to get a density estimator which outperforms any individual one. *Ensembling* diffusion models that "specialize" in different SNR ranges can likely be used to construct new SOTA density models.

For the following experiments we use the CIFAR-10 dataset, scaled to have pixel values in $[-1, 1]$, and consider a pre-trained DDPM model (Ho et al., 2020) and an Improved DDPM we refer to as IDDPM (Nichol & Dhariwal, 2021). We use 4000 diffusion steps for calculating variational bounds. For our IT based methods, the comparable parameter is how many log SNR samples, $\alpha \sim q(\alpha)$, we use for evaluation. For continuous density estimation 100 points were sufficient, while for the discrete estimator we used 1000 points (as discussed in App. B.5). More details on models and training can be found in App. B.3.

**Continuous Density Estimation with Diffusion**     We first explore the results of continuous probability density estimation on test data using variational bounds versus our approach. Recent diffusion based models treat the data as discrete, bounding $\log P(\boldsymbol{x})$, however, we could also use diffusion models to give continuous density estimates by interpreting the last denoising step as providing a Gaussian distribution over $\boldsymbol{x}$. For comparison, we can also take the variational bound for discrete data and create a density estimate by making density uniform within each bin. The results are shown in Fig. 2. For our estimator, the shaded integral between the Gaussian MMSE and the denoising MSE is used, and this provides noticeably improved density estimates. Note that for our bound calculated using the IDDPM model, we throw away the part of their network that estimates the covariance and still achieve improved results, validating our assertion that variational modeling of covariance is unnecessary.

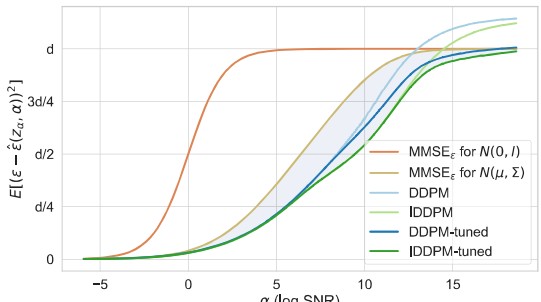

Table 1: $\mathbb{E}[-\log p(\boldsymbol{x})]$ on Test Data (bpd)

| Model | Variational Bounds | | IT |
| | Disc.* | Cont. | Cont. |
| --- | --- | --- | --- |
| DDPM | **-3.62** | -3.44 | -3.56 |
| IDDPM | -4.05 | -3.58 | **-4.09** |
| DDPM(tune) | -3.55 | -3.51 | **-3.84** |
| IDDPM(tune) | -3.85 | -3.55 | **-4.28** |
| Ensemble | - | - | **-4.29** |

Figure 2: (Left) MSE curves for different denoisers, used in estimating Negative Log Likelihood (NLL). (Right) Continuous NLL estimates for diffusion models using variational bounds and Information-Theoretic (IT) bounds (ours). Variance estimates are shown in App. B.5. *Uses a discrete estimator made continuous by assuming uniform density in each bin.

Continuous density estimators applied to intrinsically discrete pixel data, like CIFAR-10, can in principle lead to very low NLLs if the model learns to put large mass on discrete points. In our case, we are comparing the same model so the comparison would still be meaningful. Furthermore, we can see that the diffusion models are not putting probability mass on discrete points by comparing the MSE curves to denoisers where discretization is purposely introduced in Fig. 3.

**Discrete Probability**  NLL bounds for diffusion models in recent work are for discrete probability estimates. We compare to our estimate based on an exact relation between MMSE and discrete likelihood, Eqs. (12) and (16), and to our continuous NLL estimator treated as a discrete estimator with uniform dequantization. The diffusion architectures we tried do not naturally concentrate predictions on grid points at high SNR, so we had to explicitly add rounding via a "soft discretization" function (described in App. B.4), which leads to much lower error at high SNR, as seen in Fig. 3. Further improvements from ensembling the best models at different SNRs are described below.

**Fine-tuning**  Fig. 2 and 3 show that we can improve log likelihoods by fine-tuning existing models using our regression objective derived in Eq. 15, rather than the variational bound. In particular, we see that the improvements for continuous density estimation come from reducing error at high SNR levels. The inability of existing architectures to exploit discreteness in the data is the reason fine-tuning improves the continuous but not the discrete estimator in Table 2. The final discrete estimates are improved by including a soft discretization nonlinearity and using ensembling, as described next. See App. B.3 for additional experimental details.

**Ensembling**  Finally, we propose to ensemble different denoising diffusion models by choosing the denoiser with the lowest MSE at each SNR level in Eq. (14). Since our estimator depends only on estimating the smallest MSE at each $\gamma$, our likelihood estimates will benefit from combining models whose relative performance differs across SNR regions. In Fig. 2 and 3, we report improved results from ensembling DDPM (Ho et al., 2020), IDDPM (Nichol & Dhariwal, 2021), fine-tuned versions, and rounded versions (in the discrete case). For discrete estimators, rounding is useful at high SNR but counterproductive at low SNR. The ensemble MSE (shaded region of Fig. 3) gives a better NLL bound than any individual estimator.

# 6  RELATED WORK

The developments in diffusion models that have enabled new industrial applications (Ramesh et al., 2022; Saharia et al., 2022; Rombach et al., 2022) build on a rich intellectual history of ideas spanning many fields including score matching(Hyvärinen & Dayan, 2005), denoising autoencoders (Vincent, 2011), nonequilibrium thermodynamics (Sohl-Dickstein et al., 2015), neural ODEs (Chen et al., 2018), and stochastic differential equations (Song et al., 2020; Huang et al., 2021). The observation that part of the variational diffusion loss could be written as an integral of MMSEs was made by Kingma et al. (2021). To the best of our knowledge, the connection between I-MMSE relations in information theory and diffusion models has so far gone unrecognized.

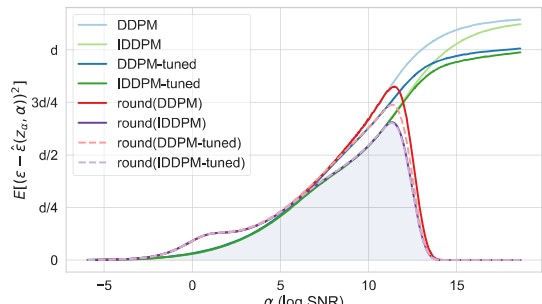

Table 2: $\mathbb{E}[-\log P(\boldsymbol{x})]$ on Test Data (bpd)

| | Var. | IT | |
|---|---|---|---|
| Model | Disc. | Disc.* | Cont.** |
| DDPM | **3.37** | 3.68 | 3.51 |
| IDDPM | **2.94** | 3.16 | 3.17 |
| DDPM(tune) | 3.45 | 3.48 | **3.41** |
| IDDPM(tune) | **3.14** | 3.15 | 3.18 |
| Ensemble | - | **2.90** | 3.16 |

Figure 3: (Left) MSE curves for different denoisers, used in estimating Negative Log Likelihood (NLL), shown with or without rounding using a soft discretization function (App. B.4). (Right) Discrete NLL estimates for diffusion models using variational bounds and information-theoretic bounds (ours). Variance estimates are shown in App. B.5. *Using the denoiser with soft discretization. **Indicates a continuous estimator made discrete via uniform dequantization.

Song et al. (2020) introduced a different way to use diffusion to model probability densities via differential equations. This approach requires solving a differential equation of a dynamic trajectory for each sample. Our estimate can be computed in a single step by evaluating the score model at each SNR in parallel, but the neural probability flow ODE will require thousands of sequential score function evaluations to solve. Concurrent work uses stochastic differential equations along with Girsanov's theorem, a stochastic version of the change of variables formula, to show that optimizing a regression objective is sufficient to exactly match a stochastic process to some target density (Liu et al., 2022; Ye et al., 2022). The results are used to improved sampling, but they use standard variational bounds for density estimation. Our work, in contrast, exactly relates regression to density estimation but does not address sampling. The form of their results for discrete distributions suggests a deeper connection between their work and this one.

Our work focused on density modeling, so approaches that forgo density modeling to improve image generation, for instance by doing diffusion in a latent space (Rombach et al., 2022), were not considered. Bansal et al. (2022) observe that denoising diffusion models using many types of noise can lead to good image sampling, however, our results highlight the special connection between Gaussian denoising and density modeling.

Recent work has studied applicability of diffusion to discrete and categorical variables (Austin et al., 2021; Gu et al., 2022; Chen et al., 2022). Our exact relation between denoising and discrete probability provides theoretical justification for this promising line of research. Finally, this paper gives a theoretical justification for ensembling diffusion models, to achieve the best MSE at different SNR levels. We see from concurrent work that this strategy is already being successfully employed (Balaji et al., 2022).

## 7 CONCLUSION

Variational methods are powerful, but require many choices in designing variational approximations. More complex choices for the variational distributions could lead to tighter bounds, and a fair amount of work has explored this possibility. Interestingly, though, the most successful refinements of diffusion models have led to objectives that are more similar to the denoising MSE, and the results in this paper finally make it clear that regression is all that is needed for exact probability estimation, for both continuous and discrete variables. We generalized the classic I-MMSE relation from information theory to introduce this simple, *exact* relationship between the data probability distribution and optimal denoisers, $-\log p(\boldsymbol{x}) = \frac{1}{2} \int_0^\infty \mathrm{mmse}(\boldsymbol{x}, \gamma) \, d\gamma + \text{constant terms}$. This result pares away the unnecessary ingredients in the variational approach, and the simple and evocative form of the result can inspire many improvements and generalizations to be explored in future work.

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
