# OpenReview forum: "Information-Theoretic Diffusion"
_ICLR.cc/2023/Conference — ICLR 2023 poster_

### Official Review · Reviewer_j334 · 2022-10-17

**Confidence:** 4
**Correctness:** 3
**Technical Novelty And Significance:** 3
**Empirical Novelty And Significance:** 2
**Recommendation:** 5

**Clarity, Quality, Novelty And Reproducibility:**

Clarity:
The clarity is clear but I feel the writing should be improved. This paper took me more time to read compared with other submissions due to its organization. I feel it is better to start with some background: the existing result/ preliminary knowledge of some important derivation and the problem.  And then begin to give some technical introduction to the information-theoretic approach. Also highlighting the important findings given by the theory in a short and simple sentence at the beginning of the section might be helpful.

Novelty:
overall, the novelty is sound.

Reproducibility:
No code is given and thus not accessible. But I believe it is reproducible.


**Strength And Weaknesses:**

Strength:
1. Using the information-theoretic approach to derive the object of the diffusion model seems interesting. Significantly, the finding of ELBO should be exact is interesting.

2. Discussion of the related theoretical property of the main result is complete.

Weakness:

1. The result that ELBO should be exact is actually not new. Existing works [1, 2] have also had such result based on the Girsanov theorem and some SDE analyses. This slightly decreases the impact of the result. Moreover, a discussion comparing this work and the existing works are necessary: I was wondering which result should be general.

2. The work in [1,2] also discusses the issue of training a discrete probability estimator but seems to using a different approach that avoids rounding and inexact approximation.

3. The sentence under equ (12): In that case, we should be able to return the exact, correct value for \hat{x} = x, leading to zero error with high probability. Could you elaborate on how this can be true during the sampling phase without the high probability that our model will be close to the discrete state at the end?

4. The experiment section is weak. I believe there should be some more intuitive/practical way to demonstrate the improvement: how about the change of some standard metric such as FID? Does your experiment supports the 4 findings you list in page 5?

[1] Let us Build Bridges: Understanding and Extending Diffusion Generative Models

[2] First Hitting Diffusion Models

===============================================================
After reading AC's comments.

Thanks AC for outputting extra information and pointing out that some presentations in this paper confused me on the `tightness of VLB`. Indeed, VLB is tight when we choose the optimal posterior is a standard result and I didn't notice that this paper also requires such optimal posterior.

Since this significantly reduces the soundness and novelty, I downgrade my initial score from 6 -> 5.

**Summary Of The Paper:**

This paper proposes using information theory tools to derive the diffusion model's optimization object without using the variational approximation. It turns out the ELBO lower bound actually matches. This paper also discusses the problem of discrete probability estimator. Experiments are done to verify the findings.

**Summary Of The Review:**

Please see above.

---

> ### Author Response · Authors · 2022-11-11
> **I-MMSE and Girsanov**
>
> Thank you for bringing this fascinating and very recent related work ([1,2]) to our attention, we will add a discussion of it. Our paper and [1,2] are quite complementary in their contributions. [1,2] focus on *exact sampling* via continuous, stochastic dynamics while our paper focuses on *exact probability estimation* (without ever invoking any dynamics).
>
> The results of [1,2,4,5] use Girsanov’s theorem to show conditions for which we can expect sampling a continuous stochastic process to perfectly match a target density. Unsurprisingly (from the point of view of our paper), the result requires only optimizing a regression objective, Eq. 4 of [1]. However, note that for empirical log likelihood estimation in [1,2], the authors used the standard discrete-time ELBO  (as described in App. B from [1]). That variational bound is sub-optimal, so the results in [1,2] would benefit by using our NLL estimator, which better matches their objective. It may be possible to derive an expression like ours from their result, but there are some nontrivial challenges that in our case are solved by exploiting results from I-MMSE theory. For instance, the undefined "constants" in Eq. 4 of [1] are not needed for optimization but are critical for density estimation. In the continuous density case, these constants can diverge, so we have to be careful to look at integrals of differences of MMSEs. These considerations are not investigated in [1,2], as they are unnecessary for training a sampler.
>
> Papers like [1,2,4,5] bolster our claim that the literature on I-MMSE theory and machine learning require a "bridge", as early results from the I-MMSE literature also show how to use Girsanov's theorem to express many relations involved in Gaussian denoising [3].
>
> > VLB is tight when we choose the optimal posterior…
>
> The top level response explains that we need to use variational distributions which are tractable but still tight, and this type of result was lacking in the literature. [2] shows that optimizing the regression objective can lead to a tight distribution match for sampling, while our paper shows why the same objective can lead to tight probability estimates. These two results together substantially answer the question of what is required to make VLBs tight and tractable - it requires continuous time / SNR and only requires a regression optimization.  No covariance modeling or special discrete decoder terms are required. These important results can help avoid wasted research effort in improving these standard but superfluous components of current diffusion models.
>
>
> [1] Let us Build Bridges: Understanding and Extending Diffusion Generative Models
>
> [2] First Hitting Diffusion Models
>
> [3] Venkat, K. and Weissman, T., 2012. Pointwise relations between information and estimation in Gaussian noise. IEEE Transactions on Information Theory, 58(10), pp.6264-6281.
>
> [4] Huang et. al, NeurIPS 2021: “A Variational Perspective on Diffusion-Based Generative Models and Score Matching”
>
> [5] Song, Durkan et. al, NeurIPS 2021: “Maximum Likelihood Training of Score-Based Diffusion Models”
>
> ---
>
> >  ... Could you elaborate on how this can be true during the sampling phase without the high probability that our model will be close to the discrete state at the end?
>
> We have no "sampling phase", as we are focused on log likelihood estimation. In the high SNR regime a (discrete) data sample falls on a discrete grid point, e.g. $x=0.1$, gets a tiny amount of noise added to get, e.g., $z = 0.10008483$. In the large SNR limit, the distance between z and x will become infinitely small, making it easy to recover the true (discrete) value of x. Some theoretical results are given in Supplementary Material, Appendix B.1.
>
> > The experiment section…
>
> Our paper is focused on probability estimation, not sampling, hence we focus on Negative Log Likelihood rather than FID. We have improved results and added some additional experiments discussing estimator variance.

---

### Official Review · Reviewer_pxZh · 2022-10-23

**Confidence:** 4
**Correctness:** 3
**Technical Novelty And Significance:** 4
**Empirical Novelty And Significance:** 3
**Recommendation:** 6

**Clarity, Quality, Novelty And Reproducibility:**

The paper is well-written and clear. The discovered connection and the technical contribution in the paper are novel and of good quality.

**Strength And Weaknesses:**

Strength:

- The paper presented a highly intriguing relationship between DPM and I-MMSE, which has the potential to have a significant impact on current research on diffusion-based generative models.
- The paper is generally well-written, organized, and easy to follow.

Questions:
- The KL terms in the original ELBO (as in DDPM) also involve MSE terms. Is there any close relation between the ELBO and the actual implementation of the proposed method (e.g. Eq.(14))? (The discussion in "Variational bound connection" seem to be not clearly enough.)
- In Section 6, it is mentioned that estimating Eq.(10) does not require a sequential estimation of the scores at different SNRs. However, I am uncertain as to whether this is an advantage over Song et al. (2020), as estimating MMSE at each SNR requires estimating an expectation over all Gaussian noise. If only one Monte Carlo sample is used to estimate each MMSE, then it is surely the proposed method is more efficient than Song et al. (2020) due to the parallelism, but this seems to produce a much larger variance. Moreover, regarding the parallelism, the original DDPM ELBO can also be evaluated parallelly if we are able to estimate the expectations (appeared in KL terms) at all SNRs. Can you please explain more about this?
- Eq.(8) looks a bit like the form of normalizing flows, does this mean that MMSE terms correspond to some kind of log determinant of Jacobian of invertible transformations?

Minors:
- In Eq.(3) the "arg" should be "argmin".


**Summary Of The Paper:**

This paper discusses an unrecognized relation between denoising diffusion probabilistic models and Information with Minimum Mean Square Error estimators (I-MMSE). By showing the equivalence of the KL divergence $KL[p(z_\gamma|x)\|p(z_\gamma)]$ and the pointwise MMSE, this work presents an exact relation between the data probability densities and the optimal denoising functions, which reveals new perspective for improving diffusion models such as recognizing important SNRs. Finally, a practical implementation of the proposed density estimator achieves good test NLLs.

**Summary Of The Review:**

Overall, this paper presents a very interesting explanation of diffusion models with I-MMSE. The technical contribution is solid.

---

> ### Author Response · Authors · 2022-11-11
> **I-MMSE / variational connections and variance**
>
> > The KL terms in the original ELBO (as in DDPM) also involve MSE terms. Is there any close relation between the ELBO and the actual implementation of the proposed method (e.g. Eq.(14))?
>
> Thank you for the question!     Indeed, in practice, both approaches implement a neural network function $\hat{\mathbf{x}}(\mathbf{z}\_{\gamma\_t}, \gamma\_t)$ that seeks to minimize the MSE:  $\mathbb{E}\_{\mathbf{x}, \mathbf{z}\_{\gamma\_t}}[ (\mathbf{x} - \hat{\mathbf{x}}(\mathbf{z}\_{\gamma\_t}, \gamma\_t) )^2] $.    Our exposition (Eq. 3, App A6) highlights that the optimum of the denoising function is the conditional expectation $\hat{\mathbf{x}}^*(\mathbf{z}\_{\gamma\_t}, \gamma\_t) = \mathbb{E}\_{\mathbf{x} | \mathbf{z}\_{\gamma\_{t}}} [ \mathbf{x} ]$.
>
> However, we agree that more thorough exposition of the connections between these objectives would improve the paper.  We have added a section in Appendix A7, which shows how discrete diffusion loss terms $\mathbb{E}\_{q(\mathbf{z}\_{\gamma\_{t}} | \mathbf{x} ) }[ D\_{KL} [ q(\mathbf{z}\_{\gamma\_{t-1}} | \mathbf{z}\_{\gamma\_{t}}, \mathbf{x} ) || p(\mathbf{z}\_{\gamma\_{t-1}} | \mathbf{z}\_{\gamma\_{t}}) ]$ from Eq. 12 of Kingma et. al [1] (or Eq. 15 for continuous time) appear in the proof of the I-MMSE relations from Guo et. al 2004.     In particular,  the I-MMSE perspective provides justification for a Gaussian approximation $p(\mathbf{z}\_{\gamma\_{t-1}} | \mathbf{z}\_{\gamma\_{t}})$ to the marginal reverse process $q(\mathbf{z}\_{\gamma\_{t-1}} | \mathbf{z}\_{\gamma\_{t}})$ in order to estimate a conditional mutual information $I(\mathbf{x}; \mathbf{z}\_{\gamma\_{t-1}} | \mathbf{z}\_{\gamma\_{t}}) $ $= \mathbb{E}\_{q(\mathbf{z}\_{\gamma\_{t}} | \mathbf{x} ) }[ D\_{KL} [ q(\mathbf{z}\_{\gamma\_{t-1}} | \mathbf{z}\_{\gamma\_{t}}, \mathbf{x} ) || q(\mathbf{z}\_{\gamma\_{t-1}} | \mathbf{z}\_{\gamma\_{t}}) ] $ $\leq \mathbb{E}\_{q(\mathbf{z}\_{\gamma\_{t}} | \mathbf{x} ) }[ D\_{KL} [ q(\mathbf{z}\_{\gamma\_{t-1}} | \mathbf{z}\_{\gamma\_{t}}, \mathbf{x} ) || p(\mathbf{z}\_{\gamma\_{t-1}} | \mathbf{z}\_{\gamma\_{t}}) ]$.       The resulting upper bound leads to the MSE terms.
>
> > If only one Monte Carlo sample is used to estimate each MMSE, then it is surely the proposed method is more efficient than Song et al. (2020) due to the parallelism, but this seems to produce a much larger variance.
>
> We find that our method is relatively low variance, using 100 samples of $(\gamma, \epsilon)$ for each $\mathbb{x}$.
>
> The revised manuscript will include variance statistics for the results in Tables 2-3, which reflect the stochasticity with respect to sampling SNR values ($\gamma$ or $\alpha$ in Eq. 15) and Gaussian noise $\epsilon$.
>
> > The original DDPM ELBO can also be evaluated parallelly if we are able to estimate the expectations (appeared in KL terms) at all SNRs
>
> We agree that the DDPM ELBO can be evaluated in parallel.   However, our experimental results show that our contributions allow us to achieve tighter bounds than the standard ELBO.

---

### Official Review · Reviewer_Ky8k · 2022-10-24

**Confidence:** 3
**Correctness:** 4
**Technical Novelty And Significance:** 3
**Empirical Novelty And Significance:** 2
**Recommendation:** 6

**Clarity, Quality, Novelty And Reproducibility:**

The paper is clear, original and good quality. The appears to contain sufficient details for reproducibility.

**Strength And Weaknesses:**

**Strengths**
- The paper is well written and establishes an interesting novel
 perspective (to the best of my knowledge) for the formalism of
 diffusion models, in terms of information and MMSE estimators.
- The method allows to refine likelihood bounds for existing models
  while being significantly more efficient in terms of required steps to compute the likelihood bounds.
- The paper proposes a principled approach to improving the bounds by
  ensembling models with the best MSE at different SNRs.

**Weaknesses/Questions**
- The resulting refined bounds seem to be mixed in terms of sometimes
  increasing or decreasing the value from the variational bounds. I
  wonder if the authors can comment further on practical implications
  that could be derived from this study, e.g. for improving diffusion
  models?
- The authors mention categorical variables in the introduction, which
  is an interesting case in light recent discrete diffusion approaches
  [Austin et al. 2021, VQ-diffusion  Gu et al 2022]. However it
  seems to me that due to the Gaussian channel model this method is
  not amenable to non-ordinal discrete variables such as VQ encodings.


**Summary Of The Paper:**

This paper presents a new theoretical perspective on diffusion models
by using an exact information-theoretical expression for the data
likelihood in terms of minimum mean squared error estimators.  This
allows to refine likelihood bounds of existing diffusion models by
considering the MSE of the denoisers at different SNRs.


**Summary Of The Review:**

This paper presents an interesting and novel theoretical perspective
into diffusion models, which form a valuable contribution.
This contribution could be stronger if there were clear practical
implications to improve diffusion models based on this perspective.

---

> ### Author Response · Authors · 2022-11-11
> **Ensembling and Non-ordinal Discrete Variables**
>
> > Strengths: The paper proposes a principled approach to improving the bounds by ensembling models with the best MSE at different SNRs.
>
> As an update on this, we noticed a paper just posted that is successfully employing ensembling of diffusion models [3].  While they do this from an architecture engineering perspective, our paper gives a theoretical justification for this approach (which allows ensembling to be combined with density estimation).
>
> > The resulting refined bounds seem to be mixed in terms of sometimes increasing or decreasing the value from the variational bounds. I wonder if the authors can comment further on practical implications that could be derived from this study, e.g. for improving diffusion models?
>
> Results for continuous density estimation using our evaluation method (Eq. 14-15) and ensembling already significantly outperform the variational lower bounds from previous work. We made some small but significant tweaks that simplify the method and improve results, and eliminate the inconsistency you mention in the discrete setting. We found that a simple Monte Carlo estimator for Eq. 15 works better than the “low discrepancy” estimator suggested by previous work (and that we used in the original submission). We were also surprised to find that the denoising network does not automatically learn to recognize the discreteness of the input signal. By adding a simple “soft discretization” layer, we were able to significantly improve results in the discrete case.
>
> > The authors mention categorical variables in the introduction, which is an interesting case in light recent discrete diffusion approaches [Austin et al. 2021, VQ-diffusion Gu et al 2022]. However it seems to me that due to the Gaussian channel model this method is not amenable to non-ordinal discrete variables such as VQ encodings.
>
> Thanks for pointing out these works, we will add the references [1,2]. Surprisingly, our results do also apply to non-ordinal discrete variables. This is something that has been discussed in the I-MMSE literature [4, Thm. 13] and we hope to investigate the practicality of this in future work.
>
> [1] Jacob Austin, Daniel D Johnson, Jonathan Ho, Daniel Tarlow, and Rianne van den Berg. Structured denoising diffusion models in discrete state-spaces.
>
> [2] Shuyang Gu, Dong Chen, Jianmin Bao, Fang Wen, Bo Zhang, Dongdong Chen, Lu Yuan, and Baining Guo. "Vector quantized diffusion model for text-to-image synthesis." CVPR 2022.
>
> [3] arXiv:2211.01324 eDiffi: Text-to-Image Diffusion Models with an Ensemble of Expert Denoisers
>
> [4] Guo et al. Mutual information and minimum mean-square error in Gaussian channels.

---

### Official Review · Reviewer_q1V3 · 2022-10-24

**Confidence:** 3
**Correctness:** 4
**Technical Novelty And Significance:** 4
**Empirical Novelty And Significance:** 2
**Recommendation:** 8

**Clarity, Quality, Novelty And Reproducibility:**

The paper is clear and well written. The presented perspective and insights seem novel and substantial.

**Strength And Weaknesses:**

STRENGTHS
- Useful extension of the classical I-MMSE relation of Guo et al.
- Exact equality between the MLL and a function of the Gaussian MMSE (for a choice of base Gaussian), which sheds light on the success of MSE in variational objectives for diffusion models.
- The above equality is derived for both continuous and discrete inputs. The latter admits a very simple equality of -log P(x) = 0.5 int_0^{infty} MMSE(x, snr) d(snr), which implies the result on Shannon entropy from Guo et al.
- The overall approach of deriving an exact relation rather than a variational objective is refreshing.

WEAKNESSES
- This is a (mostly) theory paper. There is no (substantial) empirical contribution.
- The experiments section could be more informative (e.g., the "denoising MSE objective" used for finetuning is never explicitly defined).
- Performance of the empirical bound is not addressed, though I think it's beyond the scope of the paper. It would be interesting to discuss this issue in the context of the general difficulty of entropy estimation.

**Summary Of The Paper:**

The paper characterizes the marginal log-likelihood (MLL) as an integral of the minimum mean squared error (MMSE) between the input and the optimal denoiser output under the Gaussian noise channel (integrated over all values of signal-to-noise ratio (SNR)). The main technical tools are a pointwise generalization of the I-MMSE relation and the thermodynamic integration trick. The paper applies the result to estimating data likelihood under diffusion models. Empirical versions of the MLL characterization are developed and shown to yield slightly higher test likelihoods than existing variational methods.


**Summary Of The Review:**

The paper presents a new exact relation between MLL and MMSE under Gaussian noise channels, and applies it to the likelihood estimation under diffusion models.

---

> ### Author Response · Authors · 2022-11-11
> **I-MMSE and experiments**
>
> Thank you for your comments, we are glad to see interest in the classic I-MMSE results which, in our experience, are virtually unknown in machine learning. We believe that developing these connections further can be mutually beneficial for both fields.
> > This is a (mostly) theory paper. There is no (substantial) empirical contribution.
>
> Results for continuous density estimation using our evaluation method (Eq. 14-15) and ensembling already significantly outperform the variational lower bounds from previous work. Since the initial submission, we have made small but significant tweaks which simplify the method and improve results, particularly in the discrete setting. We found that a simple Monte Carlo estimator for Eq. 15 works better than the “low discrepancy” estimator suggested by previous work (and that we used in the original submission). We were also surprised to find that the denoising network does not automatically learn to recognize the discreteness of the input signal. By adding a simple “soft discretization” layer, we were able to significantly improve results in the discrete case.
>
> > "denoising MSE objective" used for finetuning is never explicitly defined
>
> The final objective appears in Eq. 15, but we will clarify and make this more explicit in the text.
> > Performance of the empirical bound is not addressed, though I think it's beyond the scope of the paper. It would be interesting to discuss this issue in the context of the general difficulty of entropy estimation.
>
> We are adding some additional analysis of the variance of the estimator. We plan a separate work on information estimation.

---

### Author Response · Authors · 2022-11-11
**Contrast with variational methods**

We thank the reviewers and AC for their valuable comments. In this top-level comment, we will sharpen the distinction between our approach and variational methods, as this was the subject of internal discussion among the reviewers and AC. We will respond to other specific points in comments replying to each reviewer. We are posting comments now to give time for discussion, with a revised version coming in a few days.

> R4: Thanks AC for...pointing out tightness of VLB. VLB is tight when we choose the optimal posterior is a standard result and I didn't notice that this paper also requires such an optimal posterior.

We agree that exact expressions written in terms of MMSE require an optimization over the denoising function. Under some circumstances, the VLB can become tight, so we could also write an exact expression in terms of the Maximum VLB, which we could call MVLB, to formally say, $\log p(x) = MVLB(x)$. Would this exact expression be just as useful as our bound? We will discuss three reasons to prefer our result.
- What type of optimization is required? Variational optimization is harder than regression.
- When can we say the optimization is tight? VLB tightness depends on tradeoffs between tractability and expressivity in variational optimization.
- How do we handle discrete versus continuous data? VLB requires customizing the generative model and objective to the data type.


1. Regression versus variational optimization

Our paper exactly relates probability to the optimum of a mean-squared error *regression* problem.
$$-\log p(x) = 1/2 \int d\gamma ~mmse(x, \gamma) + const.$$
No variational distributions, variational bounds, Markov chains, or stochastic differential equations are used in deriving or expressing this novel result, only basic calculus. Regression problems are easy to solve with neural networks, so this is an attractive way to express probability distributions.

Variational bounds on likelihood require optimizing over *distributions* - one distribution representing a generative model, $p$, and one representing a variational approximation to the posterior, $q$.
The variational bound for diffusion (from Ho et al 2020) is written in terms of distributions specifying a forward and reverse Markov chain, $p(x\_{t-1}|x\_t), q(x\_t|x\_{t-1})$, for $t=1,\ldots,T$.
$$-\log p(x) \leq \mathbb E\_q [ D\_{KL}(q(x\_T|x\_0) || p(x\_T)) -\log p(x\_0|x\_1) +  \sum\_{t>1} D\_{KL}(q(x\_t | x\_{t-1}, x\_0 ) || p(x\_{t-1} | x\_t) ) ]  $$
Optimizing this bound over all distributions is difficult. To implement this optimization with neural networks requires making several choices to simplify the problem. By choosing diagonal Gaussians for the distributions, the third term becomes simpler, taking the form of a regression problem (more detail in response to R3), but the other terms remain and constraining the variational family affects the quality of the bound, as we discuss next.

2. Optimizing within a constrained variational family is not necessarily tight

The derivation of our result gives an exact expression and the optimality conditions are clear (see Eq. 4). For the VLB, if we allow optimization over *all possible distributions, $p,q$* then the bound can be tight, as pointed out by R4. But we saw that in practice we have to restrict our optimization to some tractable set of distributions. Is the optimization *over the subset of tractable distributions* still tight? This is a highly nontrivial question to answer.

Looking only at the second term in the VLB, for the final decoder layer $p(x\_0|x\_1)$, we see that practitioners do not expect to get tight bounds with current architectures. According to Ho et al (2020),
> It would be straightforward to instead incorporate a more powerful decoder like a conditional autoregressive model, but we leave that to future work.

Ho et al recognized that there was no guarantee of tightness for this decoder, and suggested more expressive distributions. Once again, however, our new result in Eq. 9 and 12 shows that MMSE regression is all that is required for a tight bound. (See additional detail in response to R4).

3. Continuous versus discrete objectives

Finally, note that the variational bound requires specifying a different generative model for the final decoder layer, $p(x\_0|x\_1)$, depending on whether the data is continuous or discrete. This means that the variational objective will be different for continuous and discrete data. Surprisingly, our result shows that optimizing the same objective gives tight results for both cases. This further underscores how our result is substantially different from the variational one.

---

> ### Author Response · Authors · 2022-11-15
> **Summary of changes in revised version**
>
> Below are a summary of changes in the revised version of the PDF and supplementary material.
> - Wording changes and clarifications throughout to address reviewer comments
> - Appendix A.7 added, discussing comparisons with variational bounds in detail
> - Improved results in Fig. 2 & 3, due to use of simpler Monte Carlo estimator (replacing a “low discrepancy” sampler)
> - For the discrete NLL estimator, results are now consistently improved compared to the variational bound by using a soft discretization function (B.4) and reducing variance of the integral estimate (B.5 and Sec. 5)
> - Appendix B.4 added, discussing a soft discretization function that greatly improved results for discrete NLL estimation
> - Appendix B.5 added, new results showing that our estimators have low variance

---

### Comment · Area_Chair_Li2y · 2022-11-15
**Response**

Dear reviewers,

Your response to the authors' rebuttal would be highly appreciated.

Kind regards,
Your AC

---

> ### Author Response · Authors · 2022-12-09
> **Discussion period ending**
>
> The reviewers' discussion period is ending on 12/12 but we have yet to receive any response or reaction from the reviewers. Could the Area Chair please ask the reviewers to at least acknowledge that they have read the rebuttal? We would really appreciate it!
>
> Thanks,
> Paper5479 Authors

---

### Decision · Program_Chairs · 2023-01-20

**Decision:**

Accept: poster

**Justification For Why Not Higher Score:**

Somewhat weak empirical results.

**Justification For Why Not Lower Score:**

Interesting perspective on diffusion models.

**Metareview: Summary, Strengths And Weaknesses:**

Ratings: 8/6/6/5
Confidences: 3/3/4/4

By using information-theoretical tools, the authors derive an exact expression for the data likelihood of diffusion models in terms of minimum mean squared error estimators. This allows for a more accurate estimation of density, and the proposed method also enables the simultaneous modeling of both continuous and discrete probabilities with no additional cost. Empirical results show that the proposed method yields slightly higher test likelihoods than existing variational methods.

The strengths of this paper include its novel perspective on diffusion models. The ensembling approach to improving bounds is also a strength. Weaknesses include the mixed empirical results and the limitations of the Gaussian channel model for non-ordinal discrete variables. There were various questions from reviewers, which were addressed in the rebuttal. My recommendation is to accept.

**Note From Pc:**

if the above contains the word "oral" or "spotlight" please see: "oral" presentation means -> notable-top-5% and "spotlight" means -> notable-top-25%. As stated in our emails, we are disassociating presentation type from AC recommendations